# Early Life Factors and Polycystic Ovary Syndrome in a Swedish Birth Cohort

**DOI:** 10.3390/ijerph20227083

**Published:** 2023-11-20

**Authors:** Beata Vivien Boldis, Ilona Grünberger, Agneta Cederström, Jonas Björk, Anton Nilsson, Jonas Helgertz

**Affiliations:** 1Department of Public Health Sciences, Stockholm University, 10691 Stockholm, Sweden; ilona.koupil@su.se (I.G.); agneta.cederstrom@su.se (A.C.); 2Epidemiology, Population Studies and Infrastructures (EPI@LUND), Department of Laboratory Medicine, Lund University, 22100 Lund, Sweden; jonas.bjork@med.lu.se (J.B.); anton.nilsson@med.lu.se (A.N.); 3Centre for Economic Demography, School of Economics and Management, Lund University, 22100 Lund, Sweden; jonas.helgertz@ekh.lu.se; 4Department of Economic History, Lund University, 22100 Lund, Sweden; 5Institute for Social Research and Data Innovation, Minnesota Population Center, University of Minnesota, Minneapolis, MN 55455, USA

**Keywords:** polycystic ovary syndrome, maternal diabetes, maternal smoking, developmental origins of health

## Abstract

Polycystic ovary syndrome (PCOS) is a medical condition with important consequences for women’s well-being and reproductive outcomes. Although the etiology of PCOS is not fully understood, there is increasing evidence of both genetic and environmental determinants, including development in early life. We studied a population of 977,637 singleton women born in in Sweden between 1973 and 1995, followed sometime between the age 15 and 40. The incidence of PCOS was measured using hospital register data during 2001–2012, complemented with information about the women’s, parents’ and sisters’ health and social characteristics from population and health care registers. Cox regression was used to study how PCOS is associated with intergenerational factors, and a range of early life characteristics. 11,594 women in the study sample were diagnosed with PCOS during the follow-up period. The hazard rate for PCOS was increased 3-fold (HR 2.98, 95% CI 2.43–3.64) if the index woman’s mother had been diagnosed with PCOS, and with 1.5-fold (HR 1.51, 95% CI 1.39–1.63) if their mother had diabetes mellitus. We found associations of PCOS with lower (<7) one-minute Apgar score (HR 1.19, 95% CI 1.09–1.29) and with post-term birth (HR 1.19, 95% CI 1.13–1.26). Furthermore, heavy (10+ cigarettes/day) maternal smoking (HR 1.30, 95% CI 1.18–1.44) and maternal obesity (HR 1.90, 95% CI 1.62–2.36) were strongly associated with PCOS. This study finds support for the heritability and fetal origins of PCOS. Risk of PCOS could be reduced by further emphasizing the importance of maternal and early life health.

## 1. Introduction

Polycystic ovary syndrome (PCOS), as a manifestation of ovarian dysfunction (OD), is the most common endocrine disorder in women of reproductive age, with an estimated prevalence of about 3% to 10% among women worldwide [1,2]. PCOS has been previously associated with numerous other negative health outcomes, including type II diabetes, cardiovascular disease, endometrial cancer, ovarian cancer, sleeping disorders, eating disorders, sexual dysfunction, obesity, anxiety, depression and adverse social consequences such as social isolation [3,4,5,6,7,8]. Additionally, PCOS is a major cause of subfertility and anovulation in women [9]. Yet, the etiology and natural history of PCOS is still not fully understood.

In the 10th version of the International Classification of Disease (ICD), used in Sweden from 1997, PCOS (E28.2) is included as a complex endocrine disorder with main characteristic of menstrual disturbances, hirsutism, obesity and infertility [10], and falls under the main diagnostic code of OD (E28). This is partly consistent with the generally accepted Rotterdam criteria which is widely used to diagnose PCOS [11]. Based on this, two of the following symptoms must be present in order to confirm a diagnosis of PCOS; oligo-ovulation or anovulation, hyperandrogenism and polycystic ovaries visible on ultrasound [11]. Additionally, disorders that mimic the symptoms of PCOS, such as Cushing syndrome, should be excluded [11]. Even if a uniform clinical definition of PCOS does not exist, it is commonly classified as a syndrome rather than disease. As the definition of PCOS is still debated, it is important to note that its prevalence depends on the criteria that are used for diagnosing.

It has been suggested that PCOS is caused by a combination of genetic and environmental factors [12], a process that might start in utero in a genetically predisposed fetus [12,13]. Some hypothesized mechanisms involve developmental origins and growth in fetal life [12], including a response to fetal growth restriction [14]. It has been suggested that fetal growth restriction [15], via excess serine phosphorylation [16], is associated with PCOS and insulin resistance. Furthermore, during critical periods of the pregnancy, elevated testosterone levels may also cause development of PCOS-like phenotypes [17] mediated through impaired placental function [18].

Both PCOS and insulin resistance have been linked with fetal growth restriction in previous research, [3,4,5,6,7,8]. This is manifested through clinical symptoms in puberty, affecting fertility in adult life, and continuing throughout all reproductive years of a woman’s lifespan [19]. First-degree relatives of women diagnosed with PCOS have a significantly higher likelihood to suffer from the condition compared to the general population [20], possibly suggesting a parental genetic component through autosomal dominant heritability [21]. Moreover, PCOS was shown to affect one-third of sisters of women with the condition [22]. Additionally, there is an established association between PCOS and abnormal glucose metabolism, a complex, highly heritable trait [23,24,25,26].

From a public health perspective, PCOS is of importance not merely as a condition with its related burden and suffering for the affected women, but also in terms of societal demand for reproductive health care and infertility treatments. To the best of our knowledge, the impact of both the biological and sociodemographic characteristics on the burden and etiology of PCOS has not been sufficiently studied. This, nationwide Swedish register-based cohort study, contributes by independently assessing the associations between (i) intergenerational and (ii) early life factors and polycystic ovary syndrome.

## 2. Materials and Methods

### 2.1. Study Population

We used data from the Swedish Interdisciplinary Panel (SIP) administered at the Centre for Economic Demography at Lund University. The SIP covers full birth cohorts, men and women born between 1973 and 1995, and consists of data from several Swedish administrative population and health registers. The baseline study population consisted of all 1,175,072 females born in Sweden belonging to these cohorts (index women). Through the multigenerational register we linked the index women to their parents and siblings, also if born outside of the main sampling window. We excluded multiple births and the index women without a recorded biological mother or father (40,079). Women who died or emigrated before the start of the follow-up period or before turning 15 years of age (39,980) and those with no record from the Medical Birth Register (MBR), that provided information on mothers and their offsprings’ prenatal and perinatal period since 1973, were also excluded. After additionally excluding cases with missing data on any of the explanatory variables (103,374), and women with recorded diagnoses in line with exclusion criteria (3067), the final study population consisted of 977,637 women. Figure 1 illustrates the creation of the analytical sample.

### 2.2. Construction of the Outcome Variable

The outcome of interest is being diagnosed with PCOS sometime between the age of 15 and 40, based on information obtained from the Swedish National Patient Register (NPR) [27]. Both inpatient and outpatient data from all public and private (in case of inpatient care) hospitals is covered by the NPR. The diagnosis of PCOS mostly came from outpatient data (99.7%). Due to data limitations, we were unable to a identify PCOS diagnosis (ICD-10: E28.2) and used the aggregated ICD-10 code E28. Women between the age of 15 and 40 were followed from 1 January 2001 or immigration, whatever happened later, whereas those not yet 15 years of age by the beginning of 2001 were followed from whatever happened later out of immigration or turning 15 years of age. Study subjects were followed until what happened first out of turning 40 years of age, being diagnosed with PCOS, death, emigration or the end of follow-up on 31 December 2012.

### 2.3. Exclusion Criteria

PCOS should only be diagnosed after excluding other diseases and conditions that may cause similar symptoms [11]. Based on previously used criteria for definition of PCOS in Nordic register data [28,29] and the level of detail available in our material, we used the following conditions as exclusion criteria in our analysis: Turner syndrome (Q96), malignant neoplasm of ovary (C56), suprarenal tumor (C74), adrenogenital syndrome (E25), Cushing disease (E24) and pituitary hypersecretion (E22). Index women were only included in the analysis if they have never been diagnosed with any of the exclusion criteria, regardless of whether this happened before or after the PCOS diagnosis.

### 2.4. Explanatory Variables

#### 2.4.1. Intergenerational Sociodemographic Variables

##### Mother’s Age at Index Woman’s Birth

Mother’s age at index woman’s birth was grouped to as follows: less than or equal to 18 years, between 19 and 35 years; and greater than 36 years.

##### Birth Order

Birth order was based on the mother’s live births and grouped as First born, Second born and Third born or higher.

##### Mother’s and Father’s Educational Attainment

Educational attainment (mother/father) was retrieved from the Education Register (UREG) available from 1985, but also containing retrospective information. The highest observed educational attainment was recorded and grouped into Primary, Secondary or University education.

##### Mother’s and Father’s Country of Birth

Information on country of birth (mother/father) was obtained from the Total Population Register (TPR). Countries were categorized to seven groups: Sweden; Nordics, excluding Sweden; Europe, North America and Oceania; Africa, Asia and South America.

##### Mother’s and Father’s Lifetime Earning Rank

The lifetime earnings rank for both parents was created by using annual information on taxable labor income to calculate the mother’s and father’s lifetime earnings rank from the income and taxation register. Taxable income contains taxable work-related incomes, most importantly pre-tax labor earnings, pension income and unemployment benefits. The information is available for the time-period of 1968–2011 and for each year, we calculate each individual’s earnings rank, conditional on their sex and year of birth. We restrict the calculations to observations within the ages 30–55 and obtain each individual’s annual earnings rank compared to all individuals in SIP, conditional on having the same sex and year of birth. Lifetime earnings is consequently calculated as each individual’s mean rank based on aforementioned criteria and ranging between zero and 100.

#### 2.4.2. Intergenerational Biological Variables

Mother’s and sister’s PCOS data were obtained from the NPR and classified consistently with the outcome variable meaning that the same diagnoses codes were used throughout the analysis, additionally the ICD-9 code 256 was used to identify PCOS among mothers in the earlier cohorts before the introduction of ICD-10. Mother’s and sister’s PCOS diagnoses could happen either before or after the index women’s PCOS diagnoses. Maternal diabetes mellitus (DM) was extracted from the NPR according to ICD-10 codes E10 and E11 and to ICD-9 code 250. We only measured whether the diagnosis in question was ever recorded in the data, i.e., not necessarily prior to the index woman being diagnosed with PCOS.

#### 2.4.3. Early Life Factors

Information on smoking, height and weight were only available in the MBR from 1982. Birth weight, gestational age and a one minute Apgar score were available for all birth cohorts.

##### Mother’s Smoking

Mother’s smoking in early pregnancy was analyzed as a categorical variable created from the self-reported information recorded in the MBR and classified as Not smoking, Smoking 1–9 cigarettes per day, and Smoking 10 or more cigarettes/day.

##### Mother’s BMI at the Beginning of Pregnancy

Mother’s body mass index (BMI) at the beginning of the pregnancy was computed from data on the mother’s weight and height at the first antenatal care visit. BMI categories were defined as Underweight (body mass index < 18.5 kg/m^2^), Normal weight (body mass index =18.5–24.9 kg/m^2^), Overweight (body mass index = 25–29.9 kg/m^2^), or Obese (body mass index = ≥30 kg/m^2^).

##### Weight Gain during Pregnancy

Weight gain during pregnancy was calculated from mother’s weight at the first visit at the antenatal care and mother’s weight at birth. It was classified as Inadequate, Appropriate, or Excessive based on the recommendation by the American Institute of Medicine (IOM) guidelines from 2009 [30].

##### Birth Weight

Birth weight was grouped into 500 g categories (≤2499 g, 2500–2999 g, 3000 g–3499 g, 3500–3999 g, 4000–4499 g and ≥4500 g).

##### Gestational Age

Gestational age was measured in completed weeks of gestation and categorized as extremely preterm (<28 weeks), very preterm (28–32 weeks), moderate to late preterm (33–36 weeks), normal full-term birth (37–41 weeks) and post-term birth (≥42 weeks). Information on gestational age comes from self-reported first dates of the last menstrual period or from ultrasound examinations.

##### One Minute Apgar Score

One minute Apgar score, which is a well-established and standardized assessment of health signs of newborns immediately after birth [31], was categorized as 10, 9, 8, and less than or equal to 7.

#### 2.4.4. Individual Sociodemographic Factors

Individual sociodemographic factors were included as time-varying covariates to present time-dependent within individual variations for each factor.

##### Index Woman’s Educational Attainment

The index woman’s educational attainment captures the individual’s status during the previous year, where we distinguish between less than primary, primary, secondary and university level degrees. This information was obtained from UREG. We use longitudinal information on the year when the individual attained their various degrees as the primary source, but in some cases also rely on information on the individual’s highest attained education. In the latter case, the absence of information on when the degree was obtained makes it a challenge to identify the timing of changes to the individual’s educational attainment. We therefore use information on the receipt of student benefits, assigning the highest attained education to the year subsequent to the last year of student benefit receipt. Provided that the individual’s educational attainment is more advanced, additional adjustments are made by setting the timing of transition to primary school completion to age 16, with the corresponding thresholds for secondary and university school completion to ages 19 and 21, respectively.

##### Civil Status

The civil status of the index woman was based on information from the TPR and categorized by either being Married or in a registered relationship or Not married nor in a registered relationship.

##### County of Residence

The index women’s residence at the time of the follow-up, available from the TPR and was grouped into 21 counties.

### 2.5. Statistical Analysis

To assess the associations of intergenerational and early life factors with the risk with developing PCOS at age of 15 or later, Cox proportional hazards regression models were estimated with age as the underlying duration variable. Hazard ratios with 95% confidence intervals were reported. All analyses were adjusted for birth cohorts (1973–1976, 1977–1981, 1982–1986, 1987–1991, 1992–1995). Considering that data on some early life factors, including mother’s smoking status, weight gain during pregnancy and BMI at the beginning of pregnancy only were available for women born in 1982 or later, a restricted sample was created. The full sample contained all birth cohorts from 1973 to 1995 (n = 977,637), and the restricted sample included birth cohorts 1982–1995 with complete data on all variables (n = 302,638).

Minimally adjusted analyses (Model 1 (Appendix A)), i.e., only adjusting for birth cohorts, were performed separately for each variable. Model 2 was estimated separately for each group of variables as in our conceptual diagram. These groups were intergenerational sociodemographic factors, intergenerational biological factors, early life factors and individual adult life factors. Model 3 examined the set of intergenerational biological factors while simultaneously controlling for all intergenerational sociodemographic factors considered to lie earlier in time according to our conceptual diagram. Model 4 is extended with early life factors, with Models 4a and 4b separating between the influence of pregnancy complications and birth outcomes. Finally, Model 5 included the index women’s own sociodemographic characteristics. All analyses were performed on both full and restricted samples, with Models 4a and 4b only estimating for the restricted sample. Appendix A displays the modelling strategy. The overall conceptual model for the analysis, including a set of covariates at the individual and familial levels, are displayed in Appendix A.

To assess whether fixed maternal characteristics or pregnancy related factors primarily drove the associations between early life factors and the development of PCOS among offspring, within-pair and between-pair effects were calculated and tested, as previously suggested by Mann and colleagues [32]. Between–mother associations were obtained through variables measuring the mean value of a given characteristic across mothers and daughters, and within–mother associations were obtained through each individual’s deviation from the within-mother mean. Wald tests were used to test whether the within and between associations differed, indicating residual maternal level confounding when rejecting the null hypothesis. The between–mother variables are more likely to be confounded by a range of unobserved maternal factors while the within-mother variables are more exogenous. For instance, if the association between the one-minute Apgar score and PCOS is driven by maternal level confounding, the risk for later PCOS diagnosis would be similar regardless of whether the index woman’s sister was born with a lower or higher one-minute Apgar score. In the between–within analyses, explanatory variables were either binary (one-minute Apgar score and maternal smoking) or continuous (birth weight, maternal obesity and gestational age).

### 2.6. Sensitivity Analyses and Handling of Missing Data

The main analysis is restricted to individuals with non-missing information on any of the covariates included in the models (Appendix A). Should the distributions of characteristics among the included individuals differ from those in the main sample, our results may be biased. We have therefore also estimated all reported models, including observations with missing data on any covariates. This analysis employed additional N/A (not available) categories for respective variables and yielded results that were similar to those previously reported. Additionally, comparing the results obtained from the 1973–1995 full sample with the 1982–1995 restricted sample showed no important differences, neither in terms of size nor in statistical significance. Estimating all models with standard errors clustered at the individual level yielded results that were fully consistent with the main results. Birth weight was introduced to the models both as a linear and quadratic term and no significant nonlinearity was detected.

In our main analysis, we only included women who were never diagnosed with any of the exclusion criteria, regardless of whether these happened before or after the PCOS diagnosis. Analyses where we considered whether women became diagnosed with any of the exclusion criteria before or after PCOS diagnosis specifically, yielded similar results as the main analysis.

Due to missing data concerns among individuals only observed from older ages, we ran additional analyses restricting the sample to only including individuals observed from age 20 or earlier, and age 25 or earlier (Appendix A). Those analyses also yielded results quantitatively and qualitatively similar to those reported here. Proportionality in our Cox analyses was tested with log-minus-log plots on minimally adjusted models, showing only minor violations for the N/A categories, possibly driven by their small sample size.

All data management and statistical analyses were performed in Stata version 17 (StataCorp, Lakeway, TX, USA). Figure 2 was created in RStudio version 1.3.1093. This study has obtained necessary approval from the ethics authority (dnr: 2012/627, 2017/813, 2021/02630).

## 3. Results

Among 977,637 singleton women born between 1973 and 1995, 11,594 (1.2%) were diagnosed with PCOS during the follow-up time of 2001 and 2012. In total, the follow-up included 8,526,596 person years. The incidence rate of PCOS was 1.36/1000 person years in women aged between 15 and 40 years. The incidence of PCOS increased from the age of 15 with a peak incidence at the age of 27 years, where after it progressively decreased (Figure 1). The mean age of the first diagnosis of PCOS was 26 years.

Among women with a mother diagnosed with PCOS, 3.1% became diagnosed with PCOS during the follow-up period, compared to 1.2% among women with a mother with no history of the disease (LR-test *p* < 0.001). In addition, 6.1% of those who had a sister with PCOS, became diagnosed during the follow-up period, compared to those (1.1%) with a sister with no history of the disease, and to those (1.2%) without a sister (*p* < 0.001). Table 1 displays the characteristics of women born in Sweden between 1973 and 1995. Furthermore, 68% of the women had a one-minute Apgar score of 9 and 86% were born full term. The distribution or birth weight was bell shaped with an extended lower tail. Additionally, more than half of the women (52.9%) were the first born. Around 15% of the women had a mother who was a smoker during the first antenatal visit, 12% of the women’s mothers had an excessive weight gain during pregnancy and more than 90% of the study subjects were born from women between the age of 19 and 35 years.

Most of the women were born to a Swedish-born mother (89.4%) and father (88.6%), and half of those who were born to non-Swedish mothers or fathers had parents born in other Nordic countries.

### 3.1. Intergenerational Biological Variables in Relation to PCOS in Offspring

As shown in Figure 2, maternal DM, maternal PCOS and PCOS in sisters remained associated with PCOS after adjustment. The fully adjusted risk rate for PCOS was increased 2.6-fold (HR 2.61, 95% CI 2.13–3.32) (Figure 2) if the mother herself had PCOS, compared with mothers without PCOS, and the rate was increased by approximately 1.4 -fold (HR 1.38, 95% CI 1.27–1.49) (Figure 2) if their mother had DM compared with those without. PCOS in sisters increased the rate by almost 5-fold (HR 4.74, 95% CI 4.28–5.25) (Figure 2).

### 3.2. Early Life Factors and PCOS in Offspring

Maternal smoking was a strong predictor for PCOS in offspring. As Figure 2 depicts, this association remained across all models. When accounting for other early life factors, only a slight attenuation could be observed. The daughter of a heavy smoker had a 22% (HR 1.22, 95% CI 1.10–1.36) (Figure 2, Fully adjusted model) higher rate of developing PCOS compared to daughters of non-smokers. The between-mother and within-mother coefficients for maternal smoking were similar, suggesting no evidence for residual confounding and implying a possible causal effect (Table 2). Daughters of obese mothers had 68% higher rate of PCOS (HR 1.68, 95% CI 1.42–1.99) (Figure 2, Fully adjusted model), compared to mother’s with normal weight. The within-family comparison indicated a family residual confounding in the association of maternal obesity and daughter’s PCOS (Table 2).

Interestingly, birth weight was a statistically significant predictor after adjusting for gestational age, in the within mother analysis, suggesting that a higher birth weight indeed is a protective factor against PCOS (Table 2). Being born post-term increased the rate of PCOS by 19% (HR 1.19, 95% CI 1.13–1.26) (Figure 2, Minimally adjusted model) compared to full-term births, although this became weaker and non-significant after controlling for other confounding factors in the restricted sample. Finally, a positive association between lower one-minute Apgar score and the likelihood of PCOS diagnoses was detected. When investigating the robustness of birth weight, gestational age and the one-minute Apgar score associations adjusted for the intergenerational and individual sociodemographic factors, the results remained virtually unchanged. However, this association seemed to primarily be driven by unmeasured maternal factors according to the between-within model (Table 2).

## 4. Discussion

In this total-population-based cohort of 977,637 women, we assessed the association between intergenerational and early life factors and adulthood PCOS. We found associations of low one-minute Apgar score, maternal smoking, and maternal obesity with a diagnosis of PCOS. Being born to diabetic mother or to a mother with a PCOS diagnosis were also associated with higher rate of PCOS, a relationship that remained after adjusting for a range of intergenerational or individual sociodemographic factors. Women with a sister that has a PCOS diagnosis were more prone to have the disease themselves.

### 4.1. Comparison with Previous Research

Our results were in line with previous research implying that the risk of PCOS is increased among women with a first-degree relative diagnosed with DM [33,34]. Additionally, Givens et al. suggested that PCOS may be inherited through an autosomal dominant trait, passed down from both mothers and fathers [35]. Indeed, Govind et al. (1999) [21] produced evidence suggesting that this is the case, through finding higher incidence of PCOS among women with fathers and brothers with premature (<age 30) male pattern baldness [21]. Investigating “PCOS families” Kahsar-Miller et al. [22] found that 32% of the studied PCOS patients had a sister and 24% a mother with PCOS, indicative of a transmission from both parents. Yet, the inheritance of PCOS and its gene expression is complex and still debated [20]. Questions mostly emerge around the genes that are involved and the environmental factors that can contribute to their expression. Our results provide support to earlier findings on heredity of PCOS.

Birth weight is commonly used as an indicator of the intrauterine environment, and earlier studies [28,36,37] have identified birth weight as a risk factor for developing PCOS. However, the direction of the association is inconsistent. A recent meta-analysis conducted by Sadrzadeh and colleagues in 2017 [38], found that studies reporting positive association between PCOS and birth weight are overrepresented in the literature while those which found negative or no associations remain unpublished. Sadrzadeh and colleagues [38] also revealed that the criteria for the diagnosis of PCOS which were used make a difference in the association between PCOS and birth weight. In their meta-analysis they reported an association between low birth weight and PCOS diagnosis later in life, but only if the Rotterdam criteria was used for the diagnosis of PCOS.

The results of our family analysis were more in line with studies describing higher birth weight as a protective factor. Similarly, a Swedish study by Valgeirsdottir et al. (2018) [39] reported that being born small in comparison to gestational age was associated with the risk of developing PCOS later in adult life, and suggested that this association was possibly mediated by maternal factors such as country of birth or maternal PCOS diagnosis.

Contrary, from the Nordic context high birth weight was reported as a risk factor for PCOS in a recent Danish study by Mumm and colleagues [28]. They found that fetal macrosomia was associated with an elevated risk of developing PCOS. Another study from the UK by Cresswell et al. [37] confirmed the positive association between maternal obesity, high birth weight and prolonged gestation.

Our results were broadly in line with these findings, as we observed higher risk for later diagnosis of PCOS among women with diabetic mother. However, we only revealed a weak U-shaped association between birth weight and PCOS in our minimally adjusted model, which was also estimated with low precision. Similar to Cresswell et al. [37], we also found a positive association of PCOS with prolonged gestation and with obesity in mothers.

Additionally, our results show that daughters of heavy smoker mothers had an increased risk of developing PCOS. An earlier Swedish study by Valgeirsdottir et al. [39] reported similar results to ours regarding maternal smoking. Smoking shows a clear dose–response relationship with subsequent PCOS risk, even after adjustment for other health and socioeconomic factors and also in family analysis. Maternal smoking has previously been associated with a reduced level of the luteinizing hormone among offspring [40], which might be a direct risk factor for PCOS. Our between–within family analyses findings on smoking further strengthen the causal connotation of maternal smoking and PCOS in offspring.

While our study population are Swedish-born women, a large proportion of them have parents with an immigrant background. This allowed us to investigate the role of maternal and paternal country of birth for the onset of PCOS in their offspring. Our findings suggest that women of foreign-born mothers, especially among those born in Asia or Africa, had a higher risk for developing PCOS. This is an interesting and contrary finding compared to previous research, as PCOS has usually lower prevalence among Asian population when compared to Caucasian women [41,42]. We believe that the generally good access to health care in Sweden, also for women from less privileged families, may partly explain this inconsistency.

### 4.2. Strengths and Limitations

This Swedish study is based on total population administrative register data and investigates the associations between birth weight, gestational age and the risk of developing PCOS in adulthood, in addition to studying the burden of PCOS among different social groups. Importantly our study also includes information on the sociodemographic characteristics of the study subjects and their families, including parental country of origin. The high quality of the Swedish national registers [27] and the nationwide coverage further reduces the risk of selection bias.

Importantly, using register data reduces the risk for other potential biases such as recall or information bias. However, one limitation is that we only had access to ICD codes at a three-digit level of aggregation in both inpatient and outpatient data. Therefore, we could only study OD (E28) instead of PCOS which is the most common subcategory under E28. However, we do not think that this would pose a problem, since PCOS represents a vast majority (82%) of the diagnoses under E28. Additionally, the diagnostic criteria for PCOS changed over the study period [11], thus mothers had been most commonly diagnosed prior to this shift, whereas the majority of diagnoses observed for the main study population and their sisters occurred subsequently. Also, the outpatient specialist care was only added to the Swedish NPR from 2001, meaning that most of PCOS mothers diagnoses came from the inpatient register, most likely primarily capturing the most severe cases. This is further supported by the results of Cesta and colleagues [26] who found a lower estimated prevalence of PCOS in the Swedish NPR than estimates obtained from community-based samples [43].

Another limitation is the self-reported information on maternal smoking which was collected at the first antenatal visit and mainly reflects on the exposure before and during the first trimester rather than the whole pregnancy. Additionally, data on mother’s smoking, weight gain during pregnancy and BMI at the beginning of pregnancy were only available for a subgroup of women who were born in 1982 or later. And lastly the present study only included women born in Sweden which can limit the generalizability.

## 5. Conclusions

This study finds support for the heritability and fetal origins of polycystic ovary syndrome. Intergenerational factors, such as maternal diabetes mellitus and polycystic ovary syndrome and early life factors such as maternal smoking and obesity, may contribute to the risk of being diagnosed with polycystic ovary syndrome among offspring. Interventions such as good quality pre-conception care and education on the menace of maternal smoking and obesity may attenuate the risk for later PCOS diagnosis in the offspring. Furthermore, through raising awareness of the importance of family history and predisposing factors, women with PCOS can get better access to early diagnosis and care. Specific underlying mechanisms and other risk factors that are not shared within families may influence the development of polycystic ovary syndrome and should be investigated further.

## Figures and Tables

**Figure 1 ijerph-20-07083-f001:**
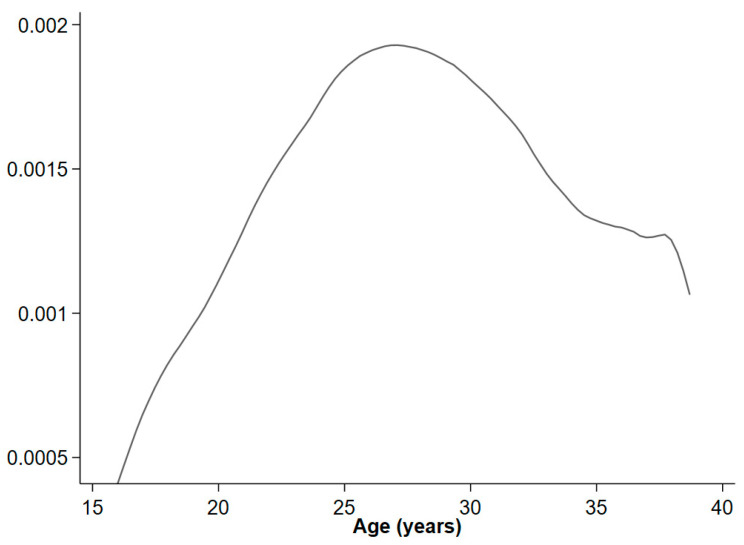
Hazard function of polycystic ovary syndrome.

**Figure 2 ijerph-20-07083-f002:**
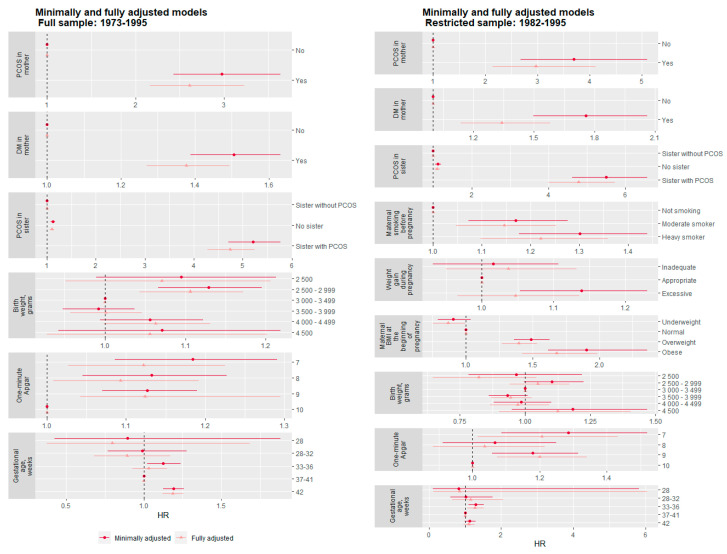
Forest plot showing the hazard ratio and 95% confidence intervals associated with intergenerational and early life factors variable throughout the different models (polycystic ovary syndrome as dependent variable).

**Table 1 ijerph-20-07083-t001:** Characteristics of index women born in Sweden between 1973 and 1995.

Characteristics	No.	%	No. of Individuals with Polycystic Ovary Syndrome	Age Adjusted Rates per 1000 (95% CI)	Share of Time at Risk (%)
Intergenerational Sociodemographic Factors					
Mother’s age at index woman’s birth, years					
Less than or equal to 18	11,962	1.19	190	1.65 [1.43–1.91]	1
Between 19 and 35	908,741	90.25	10,816	1.36 [1.33–1.38]	91
Greater than 36	86,207	8.56	998	1.40 [1.31–1.49]	8
Birth order					
First born	532,415	52.88	6904	1.42 [1.38–1.45]	55
Second born	329,102	32.68	3626	1.30 [1.25–1.34]	32
Third born or higher	145,393	14.44	1464	1.30 [1.24–1.37]	13
Mother’s educational attainment					
Primary	145,358	14.44	2072	1.52 [1.45–1.58]	16
Secondary	499,244	49.58	5935	1.37 [1.33–1.40]	49
University	359,231	35.67	3956	1.29 [1.25–1.33]	35
N/A	3077	0.31	41	1.33 [0.98–1.80]	0
Father’s educational attainment					
Primary	227,712	22.61	3107	1.47 [1.42–1.52]	24
Secondary	483,656	48.03	5560	1.35 [1.31–1.38]	47
University	285,185	28.33	3171	1.29 [1.25–1.34]	28
N/A	10,357	1.03	166	1.57 [1.35–1.83]	1
Mother’s country of birth					
Sweden	899,917	89.37	10,404	1.31 [1.29–1.34]	90
Nordics, excluding Sweden	42,581	4.23	548	1.41 [1.30–1.54]	4
Europe, North America and Oceania	36,502	3.63	596	2.00 [1.84–2.16]	3
Africa	7895	0.78	97	1.90 [1.56–2.32]	1
Asia	15,302	1.52	278	2.70 [2.40–3.04]	1
South America	4702	0.47	80	2.26 [1.81–2.81]	0
N/A	11	0.00	1	14.57 [2.05–103.42]	0
Father’s country of birth					
Sweden	892,496	88.64	10,300	1.31 [1.28–1.33]	89
Nordics, excluding Sweden	35,795	3.55	469	1.46 [1.33–1.59]	4
Europe, North America and Oceania	45,391	4.51	692	1.83 [1.70–1.97]	4
Africa	11,540	1.15	148	1.84 [1.57–2.16]	1
Asia	15,851	1.57	300	2.72 [2.42–3.04]	1
South America	5735	0.57	94	2.13 [1.74–2.60]	1
N/A	102	0.01	1	1.14 [0.16–8.11]	0
Mother’s lifetime earning rank					
First quintile	96,043	9.54	1293	1.52 [1.44–1.61]	10
Second quintile	254,925	25.32	3211	1.43 [1.38–1.48]	25
Third quintile	316,537	31.44	3671	1.33 [1.30–1.39]	31
Fourth quintile	227,909	22.63	2605	1.31 [1.28–1.37]	23
Fifth quintile	110,766	11.00	1214	1.28 [1.21–1.36]	11
N/A	730	0.07	10	1.49 [0.91–2.95]	0
Father’s lifetime earning rank					
First quintile	110,613	10.99	1561	1.64 [1.56–1.72]	11
Second quintile	245,654	24.40	3046	1.40 [1.35–1.45]	25
Third quintile	281,184	27.93	3390	1.37 [1.32–1.42]	28
Fourth quintile	223,587	22.21	2447	1.25 [1.20–1.30]	22
Fifth quintile	143,838	14.29	1534	1.25 [1.18–1.31]	14
N/A	2034	0.20	26	1.40 [0.95–2.05]	0
Intergenerational Biological Variables					
PCOS in mother					
No	1,003,708	99.68	11,907	1.36 [1.33–1.38]	100
Yes	3202	0.32	97	3.79 [3.11–4.62]	0
DM in mother					
No	973,670	96.70	11,346	1.34 [1.31–1.36]	96
Yes	33,240	3.30	658	2.07 [1.92–2.23]	4
PCOS in sister					
No sister	571,720	56.78	6819	1.38 [1.35–1.42]	56
Sister without PCOS	428,281	42.53	4768	1.25 [1.21–1.29]	43
Sister with PCOS	6909	0.69	417	6.60 [5.99–7.26]	1
Early Life Factors: Before Birth					
Mother’s smoking in early pregnancy ^a^					
Not smoking	451,750	44.86	3977	1.17 [1.13–1.21]	39
Moderate smoker	93,583	9.29	1059	1.40 [1.31–1.48]	9
Heavy smoker	56,890	5.65	757	1.62 [1.51–1.74]	5
N/A	404,687	40.19	6211	1.48 [1.45–1.52]	48
Weight gain during pregnancy ^a^					
Inadequate	88,863	8.83	1003	1.29 [1.21–1.37]	9
Appropriate	118,635	11.78	1336	1.31 [1.22–1.36]	12
Excessive	121,335	12.05	1448	1.40 [1.33–1.48]	12
N/A	678,077	67.34	8217	1.38 [1.35–1.41]	68
Mother’s BMI at the beginning of pregnancy ^a^					
Underweight	25,338	2.52	253	1.12 [1.00–1.27]	3
Normal	303,401	30.13	2888	1.20 [1.16–1.25]	27
Overweight	62,524	6.21	728	1.68 [1.56–1.81]	5
Obese	16,412	1.63	207	2.06 [1.80–2.36]	1
N/A	599,235	59.51	7928	1.40 [1.37–1.43]	64
Early Life Factors: Outcomes					
Birth weight, grams					
Less than 2500	33,599	3.34	429	1.44 [1.31–1.59]	3
2500–2999	125,625	12.48	1670	1.50 [1.42–1.57]	13
3000–3499	365,844	36.33	4301	1.33 [1.30–1.38]	37
3500–3999	339,673	33.73	3901	1.32 [1.28–1.36]	34
4000–4499	118,429	11.76	1412	1.39 [1.32–1.47]	12
Fetal macrosomia, over 4500	21,505	2.14	254	1.40 [1.24–1.58]	2
N/A	2235	0.22	37	1.81 [1.31–2.50]	0
One-minute Apgar					
Less than or equal to 7	59,002	5.86	750	1.46 [1.36–1.57]	6
8	77,786	7.73	953	1.41 [1.32–1.50]	8
9	684,433	67.97	7980	1.37 [1.34–1.40]	66
10	174,680	17.35	2184	1.28 [1.23–1.34]	19
N/A	11,009	1.09	137	1.31 [1.11–1.55]	1
Gestational age, weeks					
Extremely preterm, less than 28	861	0.09	7	1.00 [0.48–2.09]	0
Very preterm, between 2832	5949	0.59	66	1.29 [1.02–1.65]	1
Moderate to late preterm, 33–36	36,747	3.65	465	1.49 [1.36–1.63]	4
Full term, 37–41	865,841	85.99	9924	1.32 [1.30–1.35]	85
Post-term, 42 or over	94,784	9.41	1507	1.65 [1.57–1.74]	10
N/A	2728	0.27	35	1.32 [0.95–1.85]	0
Individual Adult Life Factors: Index Women					
Educational attainment					
Under primary	-	-	259	0.36 [0.32–0.40]	8
Primary school	-	-	2263	1.08 [1.04–1.13]	24
Secondary school	-	-	4 646	1.48 [1.44–1.52]	36
University	-	-	4 836	1.69 [1.65–1.74]	32
Residence at the time of follow-up					
Stockholm	-	-	3291	1.79 [1.73–1.85]	21
Uppsala	-	-	460	1.29 [1.17–1.41]	4
Södermanland	-	-	351	1.48 [1.33–1.65]	3
Östergötland	-	-	362	0.86 [0.77–0.95]	5
Jönköping	-	-	339	1.07 [0.96–1.19]	4
Kronoberg	-	-	152	0.87 [0.75–1.03]	2
Kalmar	-	-	304	1.40 [1.25–1.56]	2
Gotland	-	-	118	2.01 [1.72–2.47]	1
Blekinge	-	-	252	1.83 [1.62–2.07]	2
Skåne	-	-	1836	1.63 [1.56–1.71]	13
Halland	-	-	303	1.11 [0.99–1.24]	3
Väster Götland	-	-	1540	1.01 [0.96–1.07]	17
Värmland	-	-	251	0.99 [0.87–1.12]	3
Örebro	-	-	181	0.67 [0.58–0.77]	3
Västmanland	-	-	261	1.16 [1.01–1.28]	3
Dalarna	-	-	289	1.13 [1.01–1.27]	3
Gävleborg	-	-	407	1.61 [1.46–1.78]	3
Västernorrland	-	-	352	1.56 [1.41–1.74]	3
Jämtland	-	-	93	0.74 [0.61–0.91]	1
Västerbotten	-	-	497	1.75 [1.57–1.87]	3
Norbotten	-	-	365	1.57 [1.41–1.73]	3
Civil status					
Not married. not registered relationship	-	-	10,054	1.30 [1.27–1.32]	88
Married. registered relationship	-	-	1950	1.86 [1.78–1.95]	12

Abbreviations: BMI, body mass index; DM, diabetes mellitus; No., number; N/A, not available (missing); PCOS, polycystic ovary syndrome. ^a^ Available for a subgroup of women born between 1982 and 1995.

**Table 2 ijerph-20-07083-t002:** Comparison of between–mother and within–mother associations for characteristics independently associated with polycystic ovary syndrome and varying between siblings among Swedish women born between 1973 and 1995.

Predictors	Full Sample, 1973–1995 N = 977,637 [HR 95% CI]	Restricted Sample, 1982–1995 N = 302,638 [HR 95% CI]
One-minute Apgar score (binary: 9 or 10 vs. 8 or <7)		
Between-mother	1.079	1.050
	[1.018–1.144]	[0.942–1.171]
Within-mother	0.946	0.845
	[0.842–1.063]	[0.688–1.038]
*p*-value	0.047	0.066
Maternal smoking (binary: non-smoker vs. smoker)		
Between-mother	-	1.230
		[1.142–1.324]
Within-mother	-	1.028
		[0.723–1.462]
*p*-value	-	0.332
Maternal obesity (change per kg/m^2^ increase)		
Between-mother	-	1.059
		[1.049–1.069]
Within-mother	-	0.996
		[0.943–1.051]
*p*-value	-	0.028
Birth weight (change per kg increase)		
Between-mother	0.992	0.985
	[0.956–1.030]	[0.918–1.058]
Within-mother	0.791	0.750
	[0.716–0.875]	[0.627–0.897]
*p*-value	<0.001	0.006
Birth weight adjusted for gestational age (change per kg increase)		
Between-mother	0.959	0.985
	[0.919–1.001]	[0.908–1.066]
Within-mother	0.761	0.750
	[0.686–0.843]	[0.623–0.900]
*p*-value	<0.001	0.006

Minimally adjusted analyses adjust for the between-mother and within–mother variables in question and for daughter’s age and birth cohort. Between–mother variables represent the average across all of the mother’s offspring. Within–mother variables represent the departure of each individual cohort from the mean. *p*-values are from Wald tests for equality of between–mother and within-mother coefficients.

## Data Availability

Swedish law prohibits the distribution of sensitive information provided by the data used for this study.

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
