# Peer review of "Early Life Factors and Polycystic Ovary Syndrome in a Swedish Birth Cohort"

_ijerph, 2023, doi:10.3390/ijerph20227083_

Round 1

Reviewer 1 Report

Comments and Suggestions for Authors

The author's manuscript is full of surprises, and the statistical analysis of PCOS in this manuscript is very meaningful. After all, PCOS has become a common female disease, and since every aspect of the female reproductive system is very important to the embryo, such statistics based on real-life statistics are of great significance to the popularization of reproductive science. And there are many highlights in the manuscript that deserve to be praised.

In both the Materials and Methods, where the author's research subjects are very novel.

In line 138, the connection between PCOS and the details of the subject's family life, both in terms of bloodlines and habits, is very comprehensive and very detailed marvelous.

In line 367, the hereditary nature of PCOS is mentioned again, adding credibility by combining theory with statistical analysis of the facts.

All in all, this is a very well-developed and significant manuscript, and I very much look forward to the author's next research.

Comments on the Quality of English Language

The author's manuscript is full of surprises, and the statistical analysis of PCOS in this manuscript is very meaningful. After all, PCOS has become a common female disease, and since every aspect of the female reproductive system is very important to the embryo, such statistics based on real-life statistics are of great significance to the popularization of reproductive science. And there are many highlights in the manuscript that deserve to be praised.

In both the Materials and Methods, where the author's research subjects are very novel.

In line 138, the connection between PCOS and the details of the subject's family life, both in terms of bloodlines and habits, is very comprehensive and very detailed marvelous.

In line 367, the hereditary nature of PCOS is mentioned again, adding credibility by combining theory with statistical analysis of the facts.

All in all, this is a very well-developed and significant manuscript, and I very much look forward to the author's next research.

Author Response

We would like to thank the reviewer for their kind words and encouragement.

Reviewer 2 Report

Comments and Suggestions for Authors

This study is an effort to understand etiology and risk factors of PCOS by including 977,637 singleton women born in in Sweden between 1973 and 1995.  The risk of PCOS increased multi-folds having the mother had been diagnosed with PCOS,  diabetes mellitus, smoking, obesity, emphasizing the importance of maternal and early-life health. However it is recommended to consider the following points to revise the manuscript and resubmit. 

1. Please correlate  the mean age of the first diagnosis of PCOS to the factors and confounding factors considered in this study. 

2. Some of the facts proposed in the study are well international established. What is novel? Please explain in the introduction of the study.

3. Did you include dietary habits and lifestyle of women involved in the study?

4. Did you consider active and passive smoking both while designing the study?

5. Does the study involves any exclusion of participants based on undergoing treatment?

6. Authors are requested to explain if there is any exclusion on case of surrogacy and adoption.

7. As it is known, women with PCOS are divided into 4 groups according to their phenotypes. Did you consider these phenotypes in your study?

8. Does this sample size suffice to derive the conclusion?

9. Please refer other population based studies for better inference such as 'common risk factors among females seeking treatment for infertility in Muscat. World Journal of Pharmaceutical Research, 2014, 3(7), 79-92'

10. Please justify the statement 'Early 454 screening of women with family history of polycystic ovary syndrome and diabetes mellitus could lead to better prevention.'

Author Response

  1. Please correlate the mean age of the first diagnosis of PCOS to the factors and confounding factors considered in this study. 

Answer:

Thank you for this suggestion. We have provided a table (below) that illustrates the crude association between mean age at PCOS diagnosis and the factors controlled for in the paper. Unsurprisingly, there crude associations are consistent with the results from the main analysis. While we naturally could elaborate further on these analyses, we would like to emphasize that our analyses already account for the relationship between age at PCOS diagnosis and examined factors in a more appropriate way, through Cox proportional hazards modeling. More specifically, these models are able to account for left/right censoring/truncation as well as the influence of both time-varying and time-invariant covariates, something that neither descriptive analyses nor more standard types of regression analyses are able to do. The models presented in Table S2 (Model 1) furthermore examines the nature of the bivariate associations, similar to the table presented below.

Exposures and covariates

Mean age (years)

Mother´s age at index woman´s birth, years

Less than or equal to 18

27.61

Between 19-35

26.20

Greater than 36

24.82

Birth order

First born

27.39

Second born

25.11

Third born or higher

22.53

Mother´s educational attainment

Primary

26.76

Secondary

25.87

University

26.08

Father´s educational attainment

Primary

26.55

Secondary

25.72

University

26.17

Mother´s country of birth

Sweden

26.36

Nordics, excluding Sweden

26.65

Europe, North America and Oceania

24.70

Africa

20.65

Asia

21.56

South America

22.72

Father´s country of birth

Sweden

26.36

Nordics, excluding Sweden

26.37

Europe, North America and Oceania

25.25

Africa

22.69

Asia

21.61

South America

22.80

Mother´s lifetime earning rank

First quintile

25.56

Second quintile

25.87

Third quintile

26.13

Fourth quintile

26.41

Fifth quintile

26.54

Father´s lifetime earning rank

First quintile

25.03

Second quintile

25.99

Third quintile

26.23

Fourth quintile

26.29

Fifth quintile

26.82

PCOS in mother

No

26.14

Yes

22.59

DM in mother

No

26.06

Yes

26.93

PCOS in sister

No sister

26.51

Sister without PCOS

25.62

Sister with PCOS

25.00

Mother´s smoking in early pregnancy

Not smoking

22.28

Moderate smoker

22.17

Heavy smoker

22.10

Weight gain during pregnancy

Inadequate

23.33

Appropriate

23.08

Excessive

22.72

Mother´s BMI at the beginning of pregnancy

Underweight

23.18

Normal

22.88

Overweight

21.89

Obese

21.08

Birthweight, grams

Less than 2 500

25.58

2 500 – 2 999

26.30

3 000 – 3 499

26.05

3 500 – 3 999

26.23

4 000 – 4 499

26.01

Fetal macrosomia, over 4 500

25.45

One-minute Apgar

Less than or equal to 7

25.78

8

25.83

9

25.29

10

29.54

Gestational age, weeks

Extremely preterm, less than 28

25.71

Very preterm, between 28 - 32

25.21

Moderate to late preterm, 33 - 36

24.75

Full term, 37 - 41

25.78

Post-term, 42 or over

28.65

Educational attainment

Primary school

21.18

Secondary school

26.03

University

29.03

Civil status

Not married. not registered relationship

25.13

Married. registered relationship

30.75

  1. Some of the facts proposed in the study are well international established. What is novel? Please explain in the introduction of the study.

Answer:

We would argue that even though many of these associations are already established in some previous research, which we also cite in the paper, our paper offers a uniquely comprehensive analysis of the association between intergenerational and early-life factors and the risk of PCOS. Not only do we account for aforementioned factors more comprehensively than most previous research.

The large sample size of our study and the statistical power for more complex analysis is another advantage. Compared to earlier research in Sweden, we investigate PCOS in a sample that also includes both older and younger cohorts then in previous work and immigrants. The comprehensive investigations of the role of sociodemographic characteristics and immigration status is an original feature of our study.

In the revised manuscript, we added further justification on our study in the introduction at line 75-78 “To our best knowledge the impact of both biological and sociodemographic characteristics on the burden and etiology of PCOS has not been sufficiently studied.”

The availability of sociodemographic data on parents and the original investigation of parental country of origin are also highlighted as a strength of our study with line 428-430“Importantly our study also includes information on the sociodemographic characteristics of the study subjects and their families, including parental country of origin.”

  1. Did you include dietary habits and lifestyle of women involved in the study?

Answer:

Thank you for this suggestion. We understand why the reviewer would think dietary habits and lifestyle as an important factor to include. However, as this was a large register-based observational study, we did not have the possibility to collect information on variables that are not available in the national registries.

  1. Did you consider active and passive smoking both while designing the study?

Answer:

Thank you for this question. Ideally, it would be surely relevant to include both passive and active smoking, however similarly to the previous question; as this was a register-based study and we do not have data on passive smoking, it was impossible to control for it.

  1. Does the study involves any exclusion of participants based on undergoing treatment?

Answer:

Thank you for this question. Our observational study is focused on incidence of PCOS using the first recorded diagnosis. The treatment of PCOS was not the focus of our study.

  1. Authors are requested to explain if there is any exclusion on case of surrogacy and adoption.

Answer:

Thank you for this remark. In our analysis of complication during pregnancy, history of PCOS, maternal diabetes and family social characteristics, we use information of the study subjects and their biological mothers and fathers. In the manuscript this is explained in the Method section on line 88-89: “We excluded multiple births and index women without a recorded biological mother or father (40,079).”

  1. As it is known, women with PCOS are divided into 4 groups according to their phenotypes. Did you consider these phenotypes in your study?

Answer:

Thank you for this suggestion. We assume the reviewer refers to 2012, NIH consensus panel* proposed the phenotypic approach to classify PCOS. Phenotype A (full-blown syndrome PCOS: HA+OD+PCO) includes hyperandrogenism (HA) (clinical or biochemical), ovulatory dysfunction (OD), and polycystic ovaries (PCO) (HA+OD+PCO). Phenotype B (non-PCO PCOS: HA+OD) includes hyperandrogenism (HA) and ovulatory dysfunction (OD). Phenotype C (ovulatory PCOS: HA+PCO) includes hyperandrogenism (HA) and polycystic ovaries (PCO). Phenotype D (non-hyperandrogenic PCOS: OD+PCO) includes ovulatory dysfunction (OD) and polycystic ovaries (PCO)We agree that for better understanding of the underlying mechanisms and the natural history of PCOS, it might be helpful to make these distinctions in future clinical studies. Unfortunately, the level of detail about specific diagnoses in the register data does not allow us to classify women into these four categories or their combinations.

* Lizneva D, Suturina L, Walker W, Brakta S, Gavrilova-Jordan L, Azziz R. Criteria, prevalence, and phenotypes of polycystic ovary syndrome. Fertil Steril. 2016;106:6–15.

  1. Does this sample size suffice to derive the conclusion?

Answer:

Thank you for this question. Determining the optimal sample size of an observational study depends of course on the variability of our data, the confidence level planned to use and the effect size. We worked with a population of 977,637 singleton women, we used a 95% confidence level with an estimated prevalence of 3-10% of PCOS as previously established in the literature. In fact, the prevalence in our study was lower, however given the typical effect sizes in our research area, we believe that the population used in this study was still sufficient. As this is a nationwide study, all subjects with data available were used.

  1. Please refer other population-based studies for better inference such as 'common risk factors among females seeking treatment for infertility in Muscat. World Journal of Pharmaceutical Research, 2014, 3(7), 79-92'

Answer:

Thank you for recommending this population-based study by Al-Hassani et al., 2014. Although this is an interesting study, it is focusing on the risk factors of female Omani patients´ infertility. Our reference list is already quite extensive and we would prefer to keep to focus on papers investigating early-life and social characteristics as our research question. After considering the reviewer´s suggestion, we included two more recent studies* (line 71-72: “Additionally, there is an established association between PCOS and abnormal glucose metabolism, a complex, highly heritable trait [23]– [26].”) that supports our argument about heritability of PCOS.

* [25]              J. H. Vink, S. Sadrzadeh, C. B. Lambalk, and D. I. Boomsma, “Heritability of Polycystic Ovary Syndrome in a Dutch Twin-family study,” J. Clin. Endocrinol. Metab., vol. 91, pp. 2100–2104, 2006, doi: 10.1210/jc.2005-1494.

* [26]              C. E. Cesta, M. Månsson, C. Palm, P. Lichtenstein, A. N. Iliadou, and M. Landén, “Polycystic ovary syndrome and psychiatric disorders: Co-morbidity and heritability in a nationwide Swedish cohort,” Psychoneuroendocrinology, vol. 73, pp. 196–203, 2016, doi: 10.1016/j.psyneuen.2016.08.005.

  1. Please justify the statement 'Early screening of women with family history of polycystic ovary syndrome and diabetes mellitus could lead to better prevention.'

Answer:

We acknowledge that screening is an application of a diagnostic effort to a seemingly health population. However, this study also sheds light on those women who were born to a diabetic mother or to a mother with PCOS have a significantly higher likelihood to be diagnosed with PCOS. Based on this comment we revised the conclusion and clarified our recommendations: line 461-467 (“Interventions such as good quality pre-conception care and education on the menace of maternal smoking and obesity may attenuate the risk for later PCOS diagnosis in the offspring. Furthermore, through raising awareness of the importance of family history and predisposing factors, women with PCOS can get better access to early diagnosis and care. Specific underlying mechanisms and other risk factors that are not shared with-in-family and may influence the development of polycystic ovary syndrome should be further investigated.”)

Reviewer 3 Report

Comments and Suggestions for Authors

I appreciate the opportunity to review the manuscript entitled “Early-life Factors and Polycystic Ovary Syndrome in a Swedish Birth Cohort” submitted to journal International Journal of Environmental Research and Public Health. The authors conducted a study aimed at estimating the potential link between different demographic and clinical maternal and parental characteristics and behaviors and PCOS in daughters.

Reviewer Comments:

1.      Please prepare a list of abbreviations used in the manuscript.

2.      Please mention the potential molecular mechanisms that could explain the link between

different maternal and parental characteristics and PCOS development in daughters.

3.      Please mention whether are there animal studies about the influence of different nutrient and environmental conditions of animals on the development of PCOS in their offspring.

Taking into account the problems and results presented in above mentioned paper, my opinion is that this submission meets the criteria to be published in the journal International Journal of Environmental Research and Public Health after minor revisions and inclusion of the data I suggested.

Author Response

Reviewer Comments:

  1. Please prepare a list of abbreviations used in the manuscript.

Answer:

Thank you for this request. Please find the abbreviation list added to the manuscript.

  1. Please mention the potential molecular mechanisms that could explain the link between different maternal and parental characteristics and PCOS development in daughters.

Answer:

Thank you for this remark. As briefly discussed in our manuscript, genetics of PCOS and related comorbidities is one of the major mechanisms behind the widely documented intergenerational associations and the high heritability of PCOS. For example, Givens at al, (1988) suggests that PCOS may be inherited through an autosomal dominant trait, passed down from both mothers and fathers. In 2001, Kahsar-Miller et al. [22] studies “PCOS families” and finds evidence of transmission from both parents. Then in 2012, a genome-wide association study by Y Shi et al, identifies eight new risk loci for polycystic ovary syndrome. These and other relevant novel papers are discussed between line 368-379 in our manuscript.

Regarding the developmental origin of PCOS, commonly proposed hypothesis concerns the effect of excess serine phosphorylation, due to fetal growth restriction, on PCOS and insulin-resistance. We highlighted some of these more specific hypotheses under the introduction at line 58-63 however direct investigation of these mechanisms is not feasible in our study.

  1. Please mention whether are there animal studies about the influence of different nutrient and environmental conditions of animals on the development of PCOS in their offspring.

Answer:

      Thank you for this remark. In the absence of direct evidence on molecular mechanisms underlying the developmental origins of PCOS in humans, animal studies have been used to investigate the effect of nutrition and environment in pregnancy on health of the offspring. Intrauterine growth restriction is mediated through impaired placental function. Animal studies (Abbott et al., 2013, Padmanabhan et al., 2013)* suggest that elevated prenatal testosterone levels during critical periods of gestation can cause a PCOS like phenotype These animal studies are extremely useful to generate hypothesis about plausible biological mechanisms although direct translation of these results to interventions or treatments of human is not straightforward.

*Abbott DH, Nicol LE, Levine JE, Xu N, Goodarzi MO, Dumesic DA. Nonhuman primate models of polycystic ovary syndrome. Mol Cell Endocrinol. 2013; 373, 21–28.

*Padmanabhan V, Veiga-Lopez A. Animal models of the polycystic ovary syndrome phenotype. Steroids. 2013; 78, 734–740.
